# G protein coupling and activation of the metabotropic GABA$_B$ heterodimer

Moon Young Yang ®[1], Soo-Kyung Kim ®[1] & William A. Goddard III ®[1] ✉

Metabotropic γ-aminobutyric acid receptor (GABA$_B$R), a class C G protein-coupled receptor (GPCR) heterodimer, plays a crucial role in the central nervous system. Cryo-electron microscopy studies revealed a drastic conformational change upon activation and a unique G protein (GP) binding mode. However, little is known about the mechanism for GP coupling and activation for class C GPCRs. Here, we use molecular metadynamics computations to predict the mechanism by which the inactive GP induces conformational changes in the GABA$_B$R transmembrane domain (TMD) to form an intermediate pre-activated state. We find that the inactive GP first interacts with TM3, which further leads to the TMD rearrangement and deeper insertion of the α5 helix that causes the Gα subunit to open, releasing GDP, and forming the experimentally observed activated structure. This mechanism provides fresh insights into the mechanistic details of class C GPCRs activation expected to be useful for designing selective agonists and antagonists.

γ-Aminobutyric acid (GABA) is one of the principal inhibitory neurotransmitters in the central nervous system. It functions via the ionotropic GABA$_A$ receptor for fast responses and the metabotropic GABA$_B$ receptor (GABA$_B$R) for slow and prolonged activity[1]. GABA$_B$R is primarily coupled to the G$_{i/o}$ class of heterotrimeric G proteins (GPs), leading to a prolonged decrease in neuronal excitability via inhibition of adenylyl cyclase, voltage-gated calcium ion channels, and opening of potassium ion channels[2,3]. Because GABA$_B$R plays a central role in neurobiology, it is deeply related to various neurological diseases, making it a major pharmacological target for neurological and psychiatric disorders, including depression, schizophrenia, and addiction[4,5]. Moreover, it has been also reported that GABA$_B$R is involved in tumour development[6].

GABA$_B$R is a member of the class C G protein-coupled receptor (GPCR) family, about 20 receptors including metabotropic glutamate (mGlu), calcium-sensing, sweet taste 1, and several orphan receptors[7]. These class C GPCRs function as obligate dimers in which each subunit is composed of a transmembrane domain (TMD) 7-helix bundle and an extracellular ligand-binding domain known as the Venus flytrap (VFT) connected via a linker region. Class C GPCRs typically contain a cysteine-rich domain in the linker region, however, GABA$_B$R has a relatively short linker. Unlike other class C GPCRs such as mGlu and calcium-sensing receptors that operate as homodimers, GABA$_B$R

functions as a heterodimer with the association of two distinct subunits, GABA$_{B1}$ (GB1) and GABA$_{B2}$ (GB2)[8], although the formation of homodimer and oligomer have previously been proposed[9,10].

A crystallographic study of the VFTs showed that agonist binding to the VFT domain causes a rearrangement of the VFT and linker regions[11]. For GABA$_B$R, in particular, agonist binding occurs only in the GB1-VFT, while GP coupling and activation occur exclusively through GB2[12]. This unique allosteric mechanism for signal transduction is called trans-activation. The structure of GABA$_B$R had not been reported, except for the extracellular VFT domain, until 2019. However, recent advances in cryogenic electron microscopy (cryo-EM) have led to GABA$_B$R structures of the full-length receptor in several conformations, including both inactive and active states[9,13–16]. These structures provide valuable information for understanding the ligand binding and conformational changes upon activation in which the TMD interface between the two subunits rearranges drastically from TM3-TM5/TM3-TM5 in the inactive state to TM6/TM6 in the active state. However, this TM6/TM6 interface is hard to stabilize without the GP and/or a positive allosteric modulator (PAM). Indeed, all activated conformations of GABA$_B$R have been obtained in the presence of PAM that stabilizes the TM6/TM6 interface.

Recently, Mao et al.[17] determined the active state of GABA$_B$R complexed with G$_i$P in the presence of agonist and PAM. Consistent

[1]Materials and Process Simulation Center, California Institute of Technology, Pasadena, CA 91125, USA. ✉e-mail: wag@caltech.edu

with previous studies, it is trans-activated: G$_i$P interacts extensively with GB2, while the agonist binds to the GB1-VFT. The binding mode of the G$_i$P to the GPCR is significantly different from that of class A GPCRs. GP coupling does not involve the opening of the cytoplasmic side of TM6 (as in class A) because of hinderance by the dimeric interface. Instead, GP coupling is facilitated through coordination to intracellular loop 2 (ICL2). Similar GP binding modes have also been observed for mGlu GPCRs[18,19], indicating that this may be common for class C GPCRs. Although this agonist-GABA$_B$R-G$_i$P (with PAM) complex provides valuable information for the fully activated state, little is known about the sequence of processes (activation mechanism) by which coupling of the inactive GP with tightly bound GDP to the inactive agonist-GPCR complex induces the opening of the GP to release the GDP for exchange and signalling and finally form the final activated structure observed with cryo-EM. Moreover, in addition to the involvement of ICL2, mutation experiments showed that residues in the cytoplasmic side of TM3 play a critical role in receptor responses[17]; however, the reason has not been addressed.

We report here molecular dynamics (MD) and metadynamics (metaD) simulations to follow the sequence of mechanistic steps. MetaD is used to follow the free energy profile between states when the barrier is too high to be overcome using regular MD over a reasonable time scale. As the metaD proceeds to convergence, this process fills the potential energy for a suitably chosen set of collective variables, where the free energy surface is obtained by backing out the added terms. Using these computational techniques, we find that the G$_i$P plays an active role in the conformational change of the receptor during activation, with the C-terminus carboxylate at the end of the α5 helix of the Gα subunit (Gα5) forming an initial salt bridge (SB) with the positive TM3 residue in the inactive TMD conformation. This leads subsequently to a rearrangement of TM3-4-5 to eventually form the active TMD conformation. Then the Gα5 helix inserts further toward ICL1 simultaneous

with opening the Gα subunit to facilitate GDP release and signalling. Our proposed GP coupling mechanism provides insights into the activation mechanisms for class C GPCRs, which are quite distinct from that of class A GPCRs. We expect that this will be useful for developing new generations of agonists and antagonists that are selective for GABA$_B$R-G$_i$P.

## Results and discussion
### Structure of the activated GABA$_B$R heterodimer
Each subunit of the heterodimeric GABA$_B$R structure contains the extracellular VFT attached to the canonical TMD via the stalk domain (Fig. 1a). Both VFTs consist of two lobes (the upper and lower lobes, referred to as LB$u$ and LB$l$, respectively) with the conformation changing between open (no ligand or antagonist) and closed (with agonist). An agonist in the binding site stabilizes the closed conformation through specific interactions with residues in both LB$u$ and LB$l$. We used baclofen as the agonist for this study, an approved therapeutic drug to treat muscle spasticity and alcohol addiction[20]. The agonist binding site has SB, hydrogen bond (HB), and hydrophobic interactions with nearby residues (including S247, S270, H287, Y367, and E466) (Fig. 1b) that anchor LB$u$ and LB$l$ to maintain the GB1-VFT domain in the closed conformation (Fig. 1c). We note that the LB$u$-LB$l$ distance in the open conformation without the agonist is about 41 Å rather than 33 Å for closed.

The VFT domain is connected to the TMD through the stalk domain, which is distinct from the cysteine-rich domain observed in other class C GPCRs. This linker forms a twisted three-stranded β-sheet together with two β-strands from the long extracellular loop 2 (ECL2) of TMD. This ECL2 is nearly twice as long as for other class C GPCRs, allowing signal transduction despite the significantly short linker in GABA$_B$R. Indeed, it has been reported that ECL2 plays a crucial role in the structural transition and activation by ordering the linker connecting the extracellular VFT domain to the TMD[9].

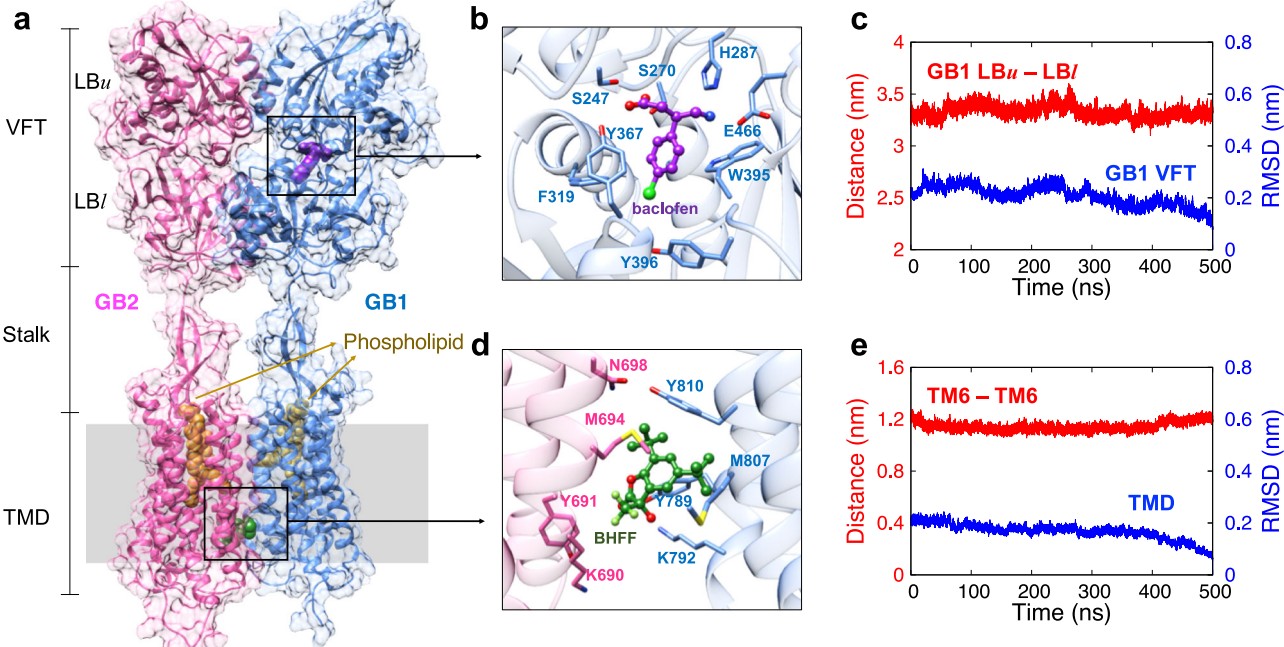

**Fig. 1 | The activated structure of GABA$_B$R heterodimer but without GP. a** The GABA$_B$R heterodimer structure contains the agonist (baclofen) attached to GB1-VFT, with the PAM (BHFF) at the GB1-GB2 TMD interface, and phospholipids (POPE) within each TMD core. **b** The agonist in the binding site anchors LB$u$ and LB$l$, keeping the closed conformation of GB1-VFT. **c** We monitored with time the distance between GB1-LB$u$ (the centre of mass of Cα for residues 222–235 and 247–260) and GB1-LB$l$ (the centre of mass of Cα for residues 347–358 and 368–382), and the GB1-VFT RMSD. **d** The PAM in the GB1-GB2 TMD interface stabilizes the TM6/TM6 interface. **e** We monitored with time the distance between the GB1-TM6 (the centre of mass of Cα for residues 802–825) and GB2-TM6 (the centre of mass of Cα for residues 689–712), and the RMSD of both TMDs. RMSDs were calculated with respect to the final frame of the MD.

Recent cryo-EM structures of the GABA$_B$R heterodimer reveal drastic conformational changes in the TMDs upon activation: a TM3-TM5/TM3-TM5 interface in the inactive state changes to a TM6/TM6 interface in the active state[13,15,16]. In fact, this TM6/TM6 interface is likely not stable and locked without the cognate GP or an allosteric partner. Thus, all known active GABA$_B$R structures contain PAM to stabilize the TM6/TM6 interface. We used BHFF as the PAM, which makes mainly hydrophobic interactions with residues in TM5 and TM6 (Y691 and M694 of GB1 and Y789, M807, and Y810 of GB2) plus an HB interaction with K792 in TM5 of GB1 (Fig. 1d). These interactions were stable over 500 ns MD simulation, maintaining the TM6-TM6 interface in the active conformation (Fig. 1e and Supplementary Fig. 1).

PAMs typically potentiate the receptor activation induced by an agonist; however, some PAMs also exhibit intrinsic agonist activity even when the agonist is not present. These are called ago-PAMs[21]. It has been reported that ago-PAMs can cause the conformational changes in GB1-VFT and TMD observed upon agonist activation[22,23]. BHFF is a known ago-PAM, which is capable of causing the receptor activity alone[23]. Since BHFF binds to the dimeric interface, it is speculated that it increases the chance for the GP engagement by stabilizing the active TM6-TM6 interface even without the agonist. In fact, it is important to develop such allosteric compounds because they can modulate the effect of orthosteric ligands to increase desirable therapeutic effects[21].

In general, both an agonist and a cognate GP are essential for activating GPCRs to induce subsequent signal transduction. Activation eventually leads to the formation of the fully activated structure, which has been characterized using cryo-EM. Although the active conformation induced solely by GP without any PAM may differ from the experimental structure obtained in the presence of PAM, many experimental results suggest that the significant conformational change to the TM6/TM6 interface in the active state arises from the nature of dimeric GPCRs[24]. Indeed, we carried out a MD simulation (800 ns) for the active structure but with PAM removed and found that the TM6/TM6 interface is stable even without PAM (at least once it is formed) (Supplementary Fig. 2).

A unique feature for GABA$_B$R is the presence of phospholipids in each of the TMD core regions, which corresponds to the orthosteric ligand binding site for class A GPCRs (Fig. 1a). These phospholipids have been observed in most cryo-EM structures of GABA$_B$R, indicating that they may contribute to stabilization and/or regulation of the TMD arrangement. However, we consider that the phospholipids are more likely structural components rather than allosteric modulators because they are observed for both inactive and active conformations. A previous MD simulation study reported that removal of the phospholipid leads to a significant decrease in the volume of the TMD cavity[9], indicating that the phospholipids are important for retaining the structure, particularly the TMD region. Based on the experimental electron density, the phospholipid is considered to be either phosphatidylethanolamine or phosphatidylcholine. Thus, we used 1-palmitoyl-2-oleoyl-sn-glycero-3-phosphatidylethanolamine (POPE) for this study. The hydrophilic head of the phospholipid interacts with the ECL2 of the TMD, while the lipid tails are buried into the deep inside the hydrophobic TMD cavity. The polar interaction between the phospholipid head and ECL2 suggests that the phospholipids may also play a role in stabilizing the linker and thus the signal transduction from the VFT domain to the TMD. Indeed, it has been shown that mutations that destabilize the interaction with the phospholipids lead to significant increase in the constitutive activity[9], indicating that the phospholipids modulate the receptor activity, presumably as a result of the linker movements.

## The structure of the GABA$_B$R–G$_i$ protein complex

The structure of the agonist-GABA$_B$R-G$_i$P complex has been reported in a recent cryo-EM study[17]. To provide a deeper understanding of the interactions between agonist, GABA$_B$R, and G$_i$P, we started from this cryo-EM structure and constructed the GABA$_B$R-G$_i$P structure containing the agonist (baclofen) and the PAM (BHFF). Then we carried out 200 ns of MD simulations to investigate the interactions between GABA$_B$R and G$_i$P (Fig. 2). In previous studies for class A GPCRs, we reported that GPCRs bind strongly to their cognate GP via SB and HB interactions at all three ICLs[25–28]. However, the G$_i$P binding mode for GABA$_B$R is significantly different than typical class A GPCRs (Supplementary Fig. 3a, b). The ICL1 of GABA$_B$R has a helical secondary structure instead of a flexible loop as in class A, and it is positioned at nearly the centre of the cytoplasmic side of the receptor. This restricts ICL1 from interacting with the Gβ subunit as observed in class A GPCRs. In addition, ICL3 is not able to make strong polar interactions with G$_i$P because TM6 is significantly shorter with no outward movement, while ICL3 does not contain any charged residues. Instead, the long ICL2 with a number of charged residues forms extensive SB interactions with the G$_i$P. During the MD simulation, we found that three positive residues in ICL2 formed stable SBs with the Gα subunit: K586−D193Gα, K588−E28GαN, and K590−D350Gα5 (started from K589) (Fig. 2b and Supplementary Fig. 4a−c). This shows that ICL2 plays a critical role in GP binding by forming multiple polar interactions with G$_i$P. Indeed, mutations of GB2-ICL2 for GABA$_B$R led to a significant reduction in receptor responses[17], and a chimera study of mGlu receptors reported that the GP coupling selectivity is determined primarily by ICL2[29]. Collectively, these results support the importance of ICL2 in GP coupling in class C GPCRs.

The insertion of the Gα5 helix deep into the intracellular region of GPCR is a pivotal step for activation[30–32]. In class A GPCRs, TM6 plays a particularly important role in this process. In the inactive state, TM6 typically forms a SB interaction to TM3 (the 'ionic lock') between R3.50 and E6.30 positions (based on the Ballesteros-Weinstein or BW numbering[33]), stabilizing the inactive conformation. Upon activation, the intracellular end of TM6 shows a significant outward movement to create a cavity on the cytoplasmic region of the receptor that accommodates insertion of the Gα5 helix more deeply into the inside of the receptor as signalling is initiated (Supplementary Fig. 3). Here, one of key interactions of the Gα5 helix is a SB from D350 to a positively charged residue in ICL2 or else to the intracellular side of TM3, both highly conserved with positively charged residues (lysine or arginine)[25–28]. Also, the terminal carboxylate of Gα5 (F354Gα5) plays an important role in positioning of the Gα5 helix by making a SB with a positively charged residue in the intracellular region of TM6.

However, class C GPCRs including GABA$_B$R have a significantly shorter TM6 compared to class A GPCRs, and the dimeric TM6/TM6 interface in the active conformation constrains outward movement of TM6 (Supplementary Fig. 3). These differences cause a significant difference in positioning the Gα5 helix, which now makes alternative interactions: F354Gα5−K513ICL1 (and K510ICL1) and D350Gα5−K590ICL2 (or K589ICL2) (Fig. 2b and Supplementary Fig. 4c, d). Similar binding modes were also observed for mGlu receptors[18,19], indicating that it is likely a common feature across class C GPCRs.

GPCR shows significant conformational change upon activation, and the movement and rotation of TM6 is a universal feature of activation across the GPCR superfamily, leading to accommodation of the Gα subunit. However, this TM6 movement displays large mechanistic differences across classes[34]. In addition to class A, it has been reported that a class B GPCR shows an significant outward movement, however in a different way; the TM6 helix is disrupted and a sharp kink is formed[35]. Also, a class D GPCR showed a unique feature, the outward movement of the extracellular side and the inward movement of the intracellular side of TM6[36]. For class F, it has been reported an outward movement of TM6 similar to class A[37]. These indicate that TM6 is a universal switch for activation in each class, however undergoing different rearrangements. Since such distinct mechanisms in each class, to understand the activation mechanism at the level of helices/residue

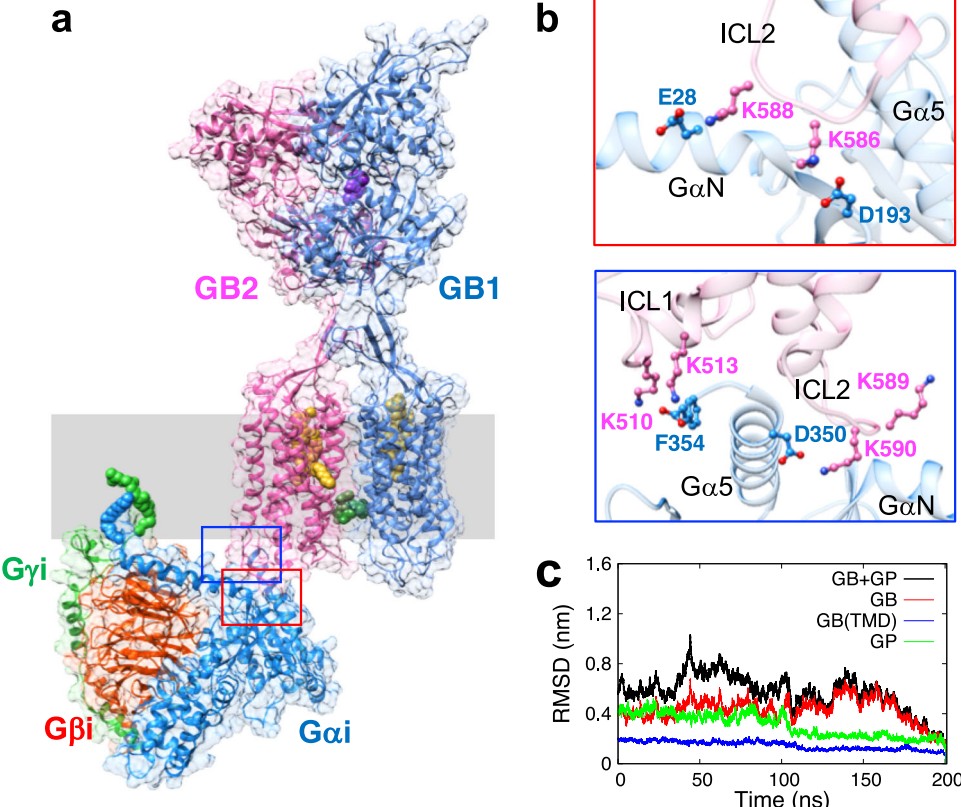

**Fig. 2 | The structure of the agonist-GABA_BR-heterotrimeric G_iP complex. a** The agonist-GABA_BR-G_iP complex exhibits trans-activation in which G_iP interacts with GB2, while the agonist binds to the GB1-VFT. **b** G_iP makes key SB interactions [K586ICL2-D193Gα, K588ICL2-E28GαN, K590ICL2-D350Gα5, and K510ICL1- F354Gα5] with GB2 at the interface. **c** The RMSD over 200 ns shows convergence of the agonist-GABA_BR-G_iP complex (calculated with respect to the final frame of the MD). Here the relatively high value for the whole protein results from fluctuations in the extracellular VFT domains connected to the flexible linker.

is important, which provides insights into rational design of experiments and, eventually, drug discovery.

## The GP coupling mechanism

For class A GPCRs, it has been proposed that stepwise sequential interactions of the Gα5 helix of the Gα subunit from its first contact to further insertion deeply inside of the cytoplasmic side of the receptor induces opening the α-helical (AH) domain of the Gα subunit and GDP release[28,38]. However, the distinct binding mode observed in the fully activated GABA_BR-G_iP complex suggests that class C GPCRs have a different GP coupling mechanism.

To understand the GP coupling mechanism, we constructed an additional GABA_BR-G_iP complex structure starting with the intermediate state of GABA_BR with agonist bound and allowing it to interact with the inactivated G_iP with GDP bound (Fig. 3a). This pre-activated state of GABA_BR was constructed based on a previously reported cryo-EM structure (intermediate-2 structure in ref. 15) that has the active TM6-TM6 interface in the TMD region, while the TM3-4-5 conformation of GB2 is still in the inactive state with no PAM in between the TMD interface (Fig. 3b, f). Although each monomer of GABA_BR shares a quite similar conformation for the inactive and active states, TM3-4-5 shows subtle but clear conformational changes upon activation. In this pre-activated structure, the GDP is tightly sandwiched between the Ras-like domain and AH domain of the Gα subunit while the Gα5 helix is in its inactive conformation, which upon activation becomes rotated and inserted deeper into the cytoplasmic side of the receptor (Fig. 3c).

Structural comparison between the inactive and active states reveals conserved 'molecular switches' that facilitate signal transduction. For class A GPCRs, one representative molecular switch is the ionic lock, a SB interaction between intracellular ends of TM3 and TM6

(typically R3.50 and E6.30 positions) that stabilizes the inactive GPCR conformation. It is well-known that breaking the ionic lock for class A allows an outward movement of the intracellular end of TM6 closer to TM5, opening TM3-6, and leading to GP engagement[39]. Although there is no D/ERY motif in TM3 for class C GPCRs, there is an alternative ionic lock between TM3 and TM6 (3.50 and 6.35 positions based on the GPCRdb numbering scheme[40]) as in class A GPCRs (Supplementary Fig. 5). It was reported for the mGlu5 receptor that mutations of these residues forming the ionic lock increases constitutive activity[41], which is consistent with the classic role of the ionic lock. For GABA_BR, mutations that abolish the ionic lock in GABA_BR resulted in a significant decrease of receptor activity[17], indicating that these residues also play an important role in the activation process, possibly beyond stabilization of the inactive state. However, class C GPCRs lack the TM6 outward movement following breaking the ionic lock, suggesting that class C GPCRs lead to a structural rearrangement to engage the GP differently from class A GPCRs. Indeed, the cryo-EM structures of inactive and active GABA_BR conformations exhibit a subtle change in the K574^3.50 and D688^6.35 residues that form the ionic lock in GB2 (Supplementary Fig. 6a). For mGlu5 receptor[41], a secondary polar lock has been proposed, involving an HB interaction between Arg3.53 and a Ser in ICL1. These residues are well conserved just as for the ionic lock (Supplementary Fig. 5), and mutations of these residues to form a stronger SB interaction instead of the original HB interaction decreased the constitutive activity[41]. This indicates that this secondary lock also plays a role in the conformational change upon activation, modulating receptor activity together with the ionic lock. Interestingly, cryo-EM structures of GABA_BR show that this secondary lock, R577^3.53 and S516^2.35, is formed in the active state, but not the inactive state (Supplementary Fig. 6b). Hence, the formation of the secondary

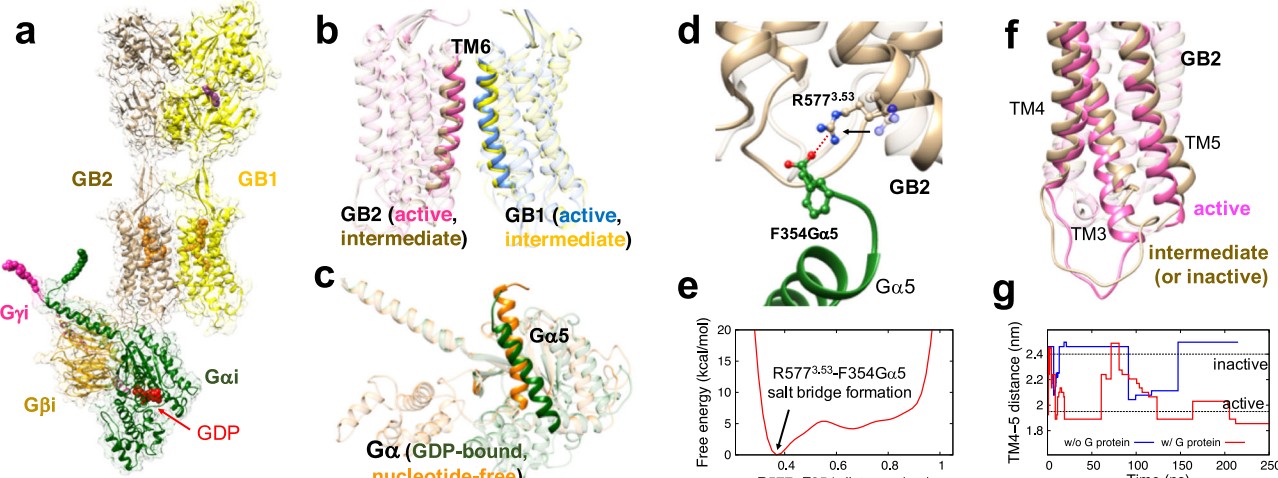

**Fig. 3 | The pre-activated agonist-GABA$_B$R-G$_i$P complex structure. a** The pre-activated structure is composed of GDP-bound inactive G$_i$P and GABA$_B$R in an intermediate state with the agonist bound to GB1-VFT but without PAM. **b** The pre-activated state of GABA$_B$R has the active TM6-TM6 interface but no PAM in between TMDs. **c** The Gα5 helix in the inactive G$_i$P with GDP bound in between Ras-like and AH domains is rotated and translated forward upon the activation. **d** As G$_i$P approaches the receptor, F354Gα5 forms a SB with GB2-R577$^{3.53}$. Here the transparent inactive conformation is superimposed for comparison. **e** The free energy analysis from metaD simulations shows that the SB formed between F354Gα5 and R577$^{3.53}$ occurs spontaneously. **f** This engagement of the G$_i$P leads to rearrangement of TM3-4-5 of GB2. **g** The TM4-5 distances (the centre of mass of Cα for residues 596–603 for TM4 and 673–680 for TM5) at the energy minima during the metaD simulations show that the G$_i$P coupling rearranges the GB2-TMD. Dotted lines show the TM4-5 distances in the cryo-EM structures, the active (1.95 nm), and the inactive (2.4 nm) states.

lock likely favours the movement of TM3 toward TM4 (or TM2) to stabilize the active conformation, opposite to the role of the normal ionic lock.

For the constructed pre-activated GABA$_B$R-G$_i$P structure, the Gα5 helix was positioned under the GB2-TM3 with F354Gα5 reaching toward the intracellular end of TM3. Of interest, F354Gα5 spontaneously forms a SB with R577$^{3.53}$ within 20 ns MD simulation (Fig. 3d). We estimated the free energy for this process with ~185 ns of metaD simulations, showing that the formation of this SB between F354Gα5 and R577$^{3.53}$ is energetically favourable (by 4.2 kcal/mol) (Fig. 3e and Supplementary Fig. 7). These results indicate that the Gα5 helix interacts first with GB2-TM3 as the GP approaches the receptor, to initiate the conformational change of the TMD. Indeed, mutations of this R577$^{3.53}$ abolished GABA$_B$R-induced production of inositol monophosphate (IP1)[17]. Moreover, mutation of I581$^{3.57}$ to tryptophan that would spatially block the TM3-F354Gα5 interaction also abolished agonist-induced receptor activity[17]. For the experimental fully activated structure, residues in TM3 including R577$^{3.53}$ do not show any direct strong interaction with G$_i$P, which cannot explain the mutation results. However, our MD results show that formation of the TM3-Gα5 interaction in the pre-activated state is the crucial step for activation, explaining the experimental mutation results. This also suggests that this pre-activated state may be able to be experimentally captured by mutating the positive residues in ICL1, although it is not clear how stable such intermediates would be. Experimental results showed that mutation of either K510A or K513A did not significantly change the receptor activity[17]. Even so, we expect that abolishing both lysine residues could trap the system in the intermediate state, preventing the complex from proceeding to the final state.

For β2-adrenergic receptor with the carboxyl terminal 14 amino acids from Gαs protein, an intermediate state was reported[42], where the charged residues of the Gα5 helix (R389 and E392) are considered to play an important role in complex formation and they change the interactions with the receptor from its first contact to the receptor to the final fully activated structure. Moreover, we reported recently for class A opioid receptors that the GDP-bound inactive G$_i$P interacts with the receptor in the inactive state to trigger conformational changes in the cytoplasmic region to form a pre-coupled GPCR-GP state prior to

agonist binding[28]. A recent cryo-EM study for mGlu2-G$_i$P also captured a state in which G$_i$P was bound to the receptor while the VFTs remained in the inactive conformation[18]. Therefore, we suggest that binding the GP to the GPCR induces a conformational change of the receptor to form the pre-activated GPCR-GP state, and this may be a common feature across GPCR families.

We find with further metaD simulations that coupling to the G$_i$P induces subsequent conformational changes of GB2-TMD. The first interaction with G$_i$P causes a downward movement of the intracellular end of TM3, which triggers further movement of the intracellular end of TM5 toward TM3, as observed in the activated conformation (Fig. 3f). Our metaD simulations show that this G$_i$P engagement drastically changes the equilibrium distance between cytoplasmic ends of TM4 and TM5 (Fig. 3g). MetaD simulations over 200 ns find converged TM4-TM5 distances of 1.86 nm with G$_i$P and 2.49 nm without G$_i$P. These distances between TM4-TM5 are close to those in cryo-EM structures of the active state (1.95 nm) and the inactive state (2.40 nm). Collectively, these results indicate that the G$_i$P plays a crucial role in the activation process by inducing the conformational change in GB2-TMD.

The deep insertion of the Gα5 helix into the intracellular region of the receptor is a crucial part of activation[28,32,38]. For GABA$_B$R, starting with the pre-activated structure, we find that the activation involves the Gα5 helix moving toward ICL1, which leads to opening the Ras-AH domain of the Gα to expose the GDP for eventual GDP-GTP exchange and signalling. Since the pre-activated structure is an intermediate state, a longer MD simulation (~μs or even longer) may lead to additional conformational changes toward the activated structure, such as the Gα opening and GDP exposure. Instead, metaD simulation can reduce the computational cost and estimate the energy along the reaction path. Thus, we performed metaD simulations to investigate the Gα opening process starting at the point at which the G$_i$P first touched the GPCR to form the initial SB between F354Gα5 and R577$^{3.53}$ where the GB2-TMD region has the activated conformation with TM3-4-5 rearranged. The predicted free energy shows that the AH domain of the Gα subunit does not yet open up when the Gα5 helix makes contact to TM3 (Fig. 4a). This is because F354Gα5 still interacts with R577$^{3.53}$ so the Gα5 helix position is not in its final position. However, after the Gα5

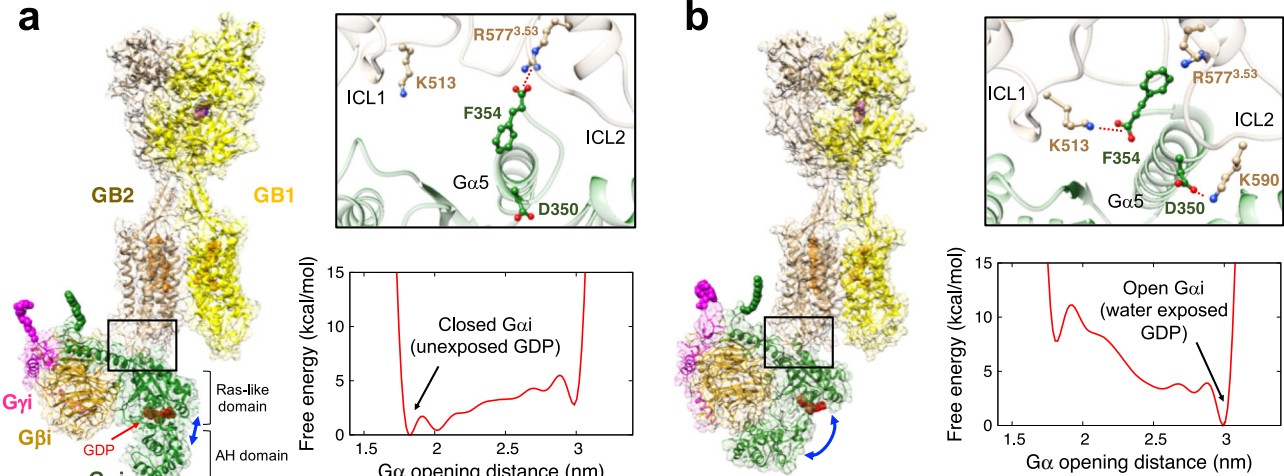

**Fig. 4 | The opening of the Gα subunit from the pre-activated agonist-GABA_BR-G_iP structure.** The structures and energy profiles starting with the pre-activated agonist-GABA_BR-G_iP structure by metaD simulations show that **a** the opening of the Gα subunit is energetically unfavourable while the Gα5 helix is in contact to TM3, **b** but opening of the Gα subunit becomes favourable when the Gα5 helix is inserted more deeply to interact with ICL1. The Gα opening distances were measured between the centre of mass of Cα for residues 44–58 in Ras-like domain, and 150–166 and 170–178 of AH domain.

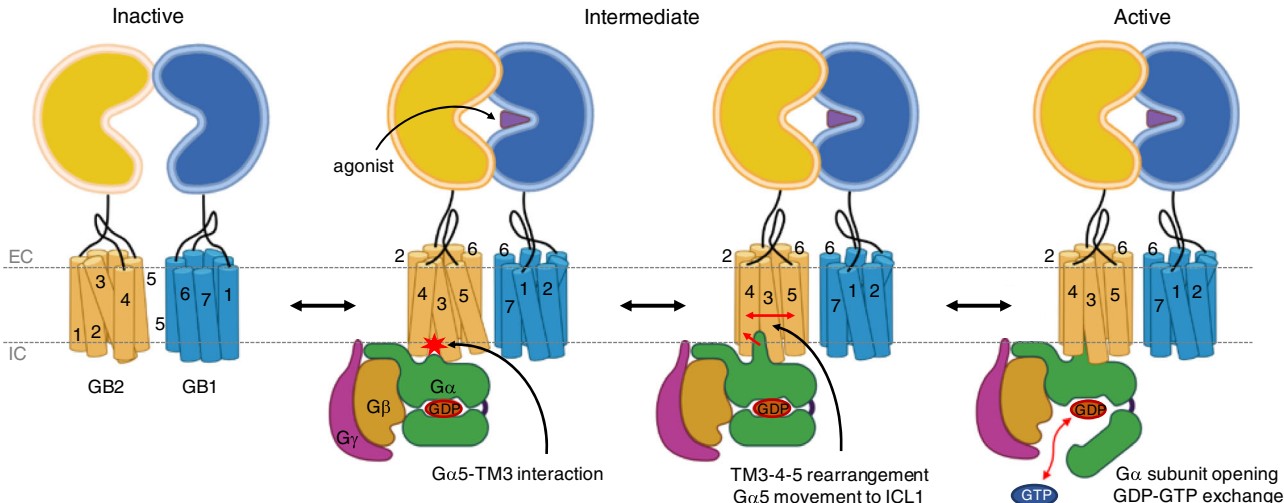

**Fig. 5 | The proposed GP coupling mechanism for GABA_BR-G_iP.** In the inactive state, both VFTs are open, and the TMDs have the TM3-5/TM3-5 interface. Agonist binding closes the GB1-VFT, leading to subsequent rearrangement of the stalk region that leads to the active TM6/TM6 interface. Then G_iP engagement initiates conformational changes of GB2-TMD by forming a SB with TM3, leading to the mutual rearrangement of TM3-4-5 in GB2. Upon the movement of the Gα5 helix toward ICL1, the Gα subunit of G_iP opens up to expose GDP to the water, allowing GDP-GTP exchange and signalling. Here EC and IC indicate extracellular and intracellular, respectively.

helix inserts deeply toward ICL1, the free energy profile for Gα opening becomes exothermic (Fig. 4b), with F354Gα5 interacting with K313ICL1 and D350Gα5 interacting with K590ICL2, as observed in our MD simulations described above for the fully activated structure (Fig. 2b). These results are consistent with previous studies showing that the Gα5 helix rotates and moves toward the cytoplasmic side of the receptor upon activation (Fig. 3c). This suggests that the Gα opening and GDP release occur after (or together with) the movement of the Gα5 helix toward ICL1 from its pre-activated state.

In summary, we carried out microseconds of MD simulations to investigate the activated GABA_BR structure with and without the G_iP. Our MD simulations show that

- The agonist interacts with residues from both LB*u* and LB*l* to maintain the closed conformation of GB1-VFT.

- The PAM at the dimeric TMDs interface stabilizes the active TM6/TM6 interface by mainly hydrophobic interactions.
- The activated TM6/TM6 interface once formed is stable even after removing PAM.
- The activated GABA_BR makes extensive SB interactions with G_iP through ICL2 of GB2, while the terminus of the Gα5 helix makes a SB with ICL1.

This G_iP binding mode is significantly different from typical class A GPCRs, which results both from the short TM6 that does not allow outward movement and from the helical secondary structure of ICL1.

We propose a mechanism for GABA_BR-GP activation based on results of microseconds of metaD simulations (Fig. 5). As the inactive G_iP approaches the agonist-GPCR complex with the inactive TMD

conformation, the negatively charged carboxylate of the Gα5 helix C-terminus (F354Gα5) makes a SB with R577[3.53], that induces the active TMD conformation by rearrangement of GB2-TM3-4-5. The Gα5 helix then inserts further into the cytoplasmic centre of the receptor to form SBs with ICL1 (and ICL2). This leads to opening the AH domain of the Gα subunit to expose the GDP for exchange with GTP and signalling. Our proposed GP coupling mechanism indicates that the GP plays an active role in the conformational changes of the receptor along a series of processes leading to activation. This provides insights into the unique activation mechanism for GABA_BR and extension to other class C GPCRs that we expect to be useful in designing agonists and antagonists with increased activity and selectivity.

## Methods

### Modelling of GABA_BR structures

To investigate the activation of the GABA_BR-G_iP complex, we prepared models for GABA_BR and GABA_BR-G_iP structures. The fully activated GABA_BR without G_iP model and with G_iP model were constructed based on the cryo-EM structures (PDB code: 7C7Q and 7EB2, respectively). For G_iP, we included the N-terminal myristoyl lipid tail of Gαi and the C-terminal geranylgeranyl lipid tail of Gγi. The pre-activated GABA_BR-G_iP structure was constructed starting from the intermediate state for the receptor (PDB code: 6UO9), where the crystal structure of the heterotrimeric G_iP bound with GDP (PDB code: 1GOT) was used as the inactive G_iP. The initial structure of the pre-activated complex structure was modelled by superimposing above structures with the equilibrated fully activated GABA_BR-G_iP structure. We used MODELLER[43] and SCWRL4[44] to add the 72 missing residues (486–493, 694–704, and 863–875 for GB1, and 295–300, 377–384, 584–594, and 749–762 for GB2, respectively) in the cryo-EM structures of the active state and the intermediate state of GABA_BR.

### System setup and equilibration

Using the CHARMM-GUI[45,46], we immersed the constructed proteins into a 1-palmitoyl-2-oleoyl-sn-glycero-3-phosphocholine (POPC) lipid bilayer, where a total of 512 and 296 POPC molecules, and 14 and 8 cholesterol molecules were used for GABA_BR with and without G_iP systems, respectively. For each model, the proteins and the lipid membrane were placed in a ~140 × 140 × 250 Å$^3$ and a ~110 × 110 × 190 Å$^3$ boxes, respectively, with water molecules and 100 mM concentration of ions (sodium and chloride). The final GABA_BR systems with and without G_iP contained ~490,000 and ~220,000 atoms, respectively.

We used the CHARMM36m force field[47], where the parameters for the ligands (agonist and PAM) were generated using the CHARMM general force field (CGenFF)[48], and the water molecules were described using the TIP3P model[49].

The constructed models were first relaxed by steepest-descent energy minimization. We then equilibrated each system by performing NVT (constant particles, constant volume, and 310 K temperature) for 200 ps followed by NPT (constant number of particles, 1 bar pressure, and 310 K temperature) for 1 ns of molecular dynamics (MD) simulation, where positional restraints were placed on the heavy atoms with an initial force constant of 9.6 kcal mol$^{-1}$ Å$^{-2}$ for proteins, agonist, PAM, and POPEs in the TMD core, and 2.4 kcal mol$^{-1}$ Å$^{-2}$ for lipids (POPC and cholesterol). These force constants were gradually reduced during the simulation to 2.4 kcal mol$^{-1}$ Å$^{-2}$ for proteins, agonist, PAM, and POPEs, and to 0 kcal mol$^{-1}$ Å$^{-2}$ for lipids. Subsequently, further 10 ns NPT simulations were performed to relax the proteins, agonist, PAM, and POPEs. Positional restraints were placed on the heavy atoms that were gradually reduced from a force constant of 2.4 kcal mol$^{-1}$ Å$^{-2}$ to 0 kcal mol$^{-1}$ Å$^{-2}$ for the first 5 ns, where the forces on the backbone atoms were kept. After 5 ns, positional restraints placed on the backbone atoms were gradually reduced from a force constant of 2.4 kcal mol$^{-1}$ Å$^{-2}$ to 0 kcal mol$^{-1}$ Å$^{-2}$ for 5 ns.

### MD and metaD simulations

All simulations were carried out using GROMACS[50] with PLUMED[51], except the case of the GABA_BR heterodimer without PAM described below. The temperature was maintained at 310 K while the pressure was controlled at 1 bar using a Parrinello–Rahman barostat[52] with a 5.0 ps damping constant, where we used semi-isotropic pressure coupling with a compressibility of 4.5 × 10$^{-5}$ bar$^{-1}$. The Lennard–Jones cutoff radius was 10 Å. The PME method[53] with a real cutoff radius of 10 Å and a grid spacing of 1.2 Å was used to calculate the long-range coulombic interactions. The P-LINCS algorithm[54] was used to keep fixed the bond lengths involving hydrogen. The time step was 2 fs and coordinates were saved every 10 ps. The total simulation times for GABA_BR heterodimer with and without G_iP were 200 and 500 ns, respectively.

The MD simulation for the GABA_BR heterodimer without PAM was performed for 800 ns on Anton2[55] in the NPT ensemble with 1 bar and 310 K. Temperature and pressure were controlled by a Nose-Hoover thermostat[56] and a semi-isotropic Martyna–Tobias–Klein barostat[57] with time constants of 0.04167 ps. The Lennard-Jones potential was used with a cutoff of 9 Å, while electrostatics were calculated using the u-series method[55]. Water molecules and bonds involving hydrogen were constrained using the M-SHAKE algorithm[58]. The time step was 2.5 fs, and coordinates were saved every 120 ps.

MetaD simulations for free energy analysis were performed using PLUMED implemented in GROMACS. The constructed pre-activated structure was first minimized and equilibrated with same procedure as described above, then a 20 ns MD simulation was performed to further equilibrate the system before starting metaD simulations. Free energy profiles as a function of simulation time were plotted to check the convergence of metaD simulations (Supplementary Figs. 7 and 8).

- For the formation of a SB between R577[3.53] and the Gα5F354, we carried out ~185 ns metaD simulation (Fig. 3e and Supplementary Fig. 7), where the bias force was applied on the distance between the C atom in the guanidine group of R577[3.53] and the C atom in the carboxylate group of Gα5F354. The bias was imposed with a Gaussian width of 0.5 Å, an initial Gaussian amplitude of 0.48 kcal/mol, a deposition period of 1.0 ps, and a bias factor of 15. To expedite the sampling process and avoid exploring undesirable regions of phase space, we imposed an upper wall on our collective variable with the force constant of ~24 kcal mol$^{-1}$ Å$^{-2}$ at distance of 9 Å.
- We performed over 200 ns of metaD simulations for the pre-activated GABA_BR-G_iP complex and the intermediate state of GABA_BR to estimate the optimum TM4-5 distances depending on the presence of G_iP (Fig. 3g). The bias forces were applied on the distance between the centre of mass of Cα's for residues 596–603 of TM4 and 673–680 of TM5. For these residues, the distance between N (i + 4) and C(i) atoms of the residues were restrained at a distance of 4.1 Å with a force constant of ~1.2 kcal mol$^{-1}$ Å$^{-2}$ to avoid deformation of the helical secondary structure by the imposed artificial forces. The bias was imposed with a Gaussian width of 1.0 Å, an initial Gaussian amplitude of 0.72 kcal/mol, a deposition period of 1.0 ps and a bias factor of 25. The lower and upper walls were imposed on the collective variable with the force constant of ~24 kcal mol$^{-1}$ Å$^{-2}$ at distances of 18.5 and 25 Å, respectively.
- The free energies for opening of the Gα5 subunit were estimated by over 200 ns metaD simulations for the pre-activated GABA_BR-G_iP (Fig. 4 and Supplementary Fig. 8). The initial structure was obtained from the previous metaD simulation with the active conformation of GB2-TM3-4-5, where the Gα5F354 was in contact to either R577[3.53] or K513ICL1 and restrained to retain the SB with a force constant of ~1.2 kcal mol$^{-1}$ Å$^{-2}$. The bias force was applied on the distance between the centre of mass of Cα for residues 44–58 in the Ras-like domain, and 150–166 and 170–178

of the AH domain. The bias was imposed with a Gaussian width of 1.0 Å, an initial Gaussian amplitude of 0.72 kcal/mol, a deposition period of 1.0 ps, and a bias factor of 25. To expedite the sampling process and avoid exploring undesirable regions of phase space, the lower and upper walls were imposed on the collective variable with the force constant of ~24 kcal mol$^{-1}$ Å$^{-2}$ at distances of 18 and 30 Å, respectively.

VMD[59] and Chimera[60] programs used for analysis and visualization.

## Reporting summary
Further information on research design is available in the Nature Research Reporting Summary linked to this article.

## Data availability
The main data supporting the findings of this study are available within the article and its Supplementary Information. The cryo-EM and X-ray structures used in this study are available in the Protein Data Bank database under accession codes 7C7Q, 7EB2, 6UO9, and 1GOT. Additional data are available from the corresponding author upon reasonable request.

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

## Acknowledgements

This research was supported by gifts to the MSC with some support from the NIH (R01HL155532). A part of the computational resource was provided by the Anton2 computer at the Pittsburgh National Supercomputing Centre (MCB180091P).

## Author contributions

M.Y.Y. and W.A.G. planned the project, M.Y.Y. carried out all computations. M.Y.Y. wrote the manuscript with help from W.A.G. and S.K.K. All authors approved the final manuscript.

## Competing interests

The authors declare no competing interests.
