## [Peer Review File · Nature Communications]

REVIEWER COMMENTS

Reviewer #1 (Remarks to the Author):

The GABABR is a class C GPCR whose active conformation has recently been solved, demonstrating a significant conformational shift from the inactive or partially activated structures. The authors used MD and metaD simulations to predict the mechanism of activation for this heterodimeric receptor. They discuss a novel mechanism of activation and opening of the G protein to exchange GDP for GTP. This may be common mechanism amongst class C GPCRs.

The authors used MD and metaD simulations to predict the conformational changes that the GABABR undergoes during activation and G protein coupling. This is the first proposed mechanism for this in Class C GPCRs and represents a significantly different mechanism from that of Class A GPCRs. The predicted mechanism explains the differences seen in the various cryo-EM structures of the GABABR in different activation states.

Major Comments:

1. The methods described in this manuscript can be used to predict the structural rearrangements of different transmembrane proteins and their signaling partners. Specifically, now that the interactions that govern G protein coupling and activation for this heterodimeric receptor have been mapped, it is possible to design novel agonists/antagonists/allosteric modifiers to trap the receptor in different activation states. It also paves the way for similar studies in other Class C GPCRs. A discussion explicitly suggesting how one could trap the different conformations experimentally based on their simulation results will be critical for impact in the field.
2. The significance of the elucidation of the meta-stable state of mGABAB is striking; however, there is no experimental evidence that validates the in silico findings. The authors could reference Liu et al 2019 (insert reference to Kobilka's β 2AR - Gs fusion protein) and perhaps replicate the results using a Gi fragment - mGABAB fusion protein. Alternatively, they could consider using hydrogen-deuterium exchange to see if there is exchange between the residues implicated in these interactions, or perhaps label specific residues and use NMR to study the meta-stable state. These maybe outside the scope of the manuscript but an explicit set of hypotheses they can deduce and discussion of experimental design that will allow others to prove or disprove the proposed mechanism will be critical for consideration.
3. As presented, there is little novelty in the MD simulations by themselves, leaving only the uniqueness of the activation mechanism. Some experimental data validating the meta-stable state will strengthen

the results. Otherwise, the study represents an important prediction and could be published as such upon appropriate revision. Irrespective of this, it will be important to discuss the findings by comparing and contrasting what has been discussed in these recent papers:

<https://pubmed.ncbi.nlm.nih.gov/35296853/>

<https://pubmed.ncbi.nlm.nih.gov/34759375/>

Other Comments

1. The text is difficult to read due to the excessive use of acronyms. If a word or phrase is used fewer than 3 times, it does not need to be abbreviated (ie, CRD, Cysteine Rich Domain is only used twice in the entire document).
2. The methods section does not state the length of time for the individual MD production runs. This information is necessary to assess the confidence in sampling sufficient conformational space within the MD simulation.
3. There are several uses of italics that seem unnecessary, pg 2 line 3, 6, 14, 36, 38; pg 3 line 16
4. The text needs careful reading, as there are places where articles (page 2 line 2: ...is one of <the> principal inhibitory...) or other words have been left out
5. Page 7 line 26 "It is well known..." the reviewer feels this statement requires a citation
6. Page 8 line 25 the sentence that begins with "Hence,..." Hence is not the appropriate word, as the statement does not flow logically from the previous sentence.
7. Page 9 line 23 "...helix rotates and translates into the cytoplasmic side..." translates does not seem like the best verb here

Reviewer #2 (Remarks to the Author):

In the present study molecular dynamics and molecular metadynamics are used to study the molecular mechanisms of Gi-protein interactions and activation of the GABA-B receptor. The entire activation process following agonist binding is not studied, but focus is on interactions between the receptor and the Gi-protein. The study is well designed and performed and necessary information is given in the methods part for reproducing the results. To my opinion, the manuscript can be accepted for publication after some changes.

Comments/suggestions to the authors:

1. A main observation from these studies was that that the Gi-protein plays an active role in the conformational changes of the receptor during activation leading to a rearrangement of TM3-4-5 of GB2 to form an active TMD conformation. However, this seems to be contradictory to the cryo-EM study by Chen and co-workers from 2021 (ref. 17 in the manuscript). They write that the cryo-EM structure of the agonist-PAM-Gi-bound GABAB structure was almost identical to that observed with the agonist and PAM without Gi, indicating that Gi had no effect on the conformation of the receptor. Baclofen and BHFF were also used in the cryo-EM study. The authors should comment on that in the results and discussion section.

2. Not alle readers are familiar with metadynamics, and the author should therefore include a short description of metadynamics in the introduction, and why it is convenient to use that for studying the coupling mechanism.

3. In the first paragraph of the introduction, they mention that GABA-B as a drug target in neurological and psychiatric disorders and give three references (4-6). Reference 6 is connected to cancer?

4. In my version, the methods part is inserted into the middle of the reference list before all references to the methods part. Something wrong with the formatting?

5. The cryo-EM structures show a PAM binding site in between the TMD interfaces, and that the PAM stabilizes an active TMD conformation. MD simulations showed that the active TMD conformation with the TM6/TM6 interface is stable without BHFF present. AgoPAMs can activate the receptor without agonist present, indicating that agoPAMs can induce active receptor conformations. Based on your simulations and available cryo-EM structures can you comment/speculate on the mechanisms of GABA-B agoPAMs.

Reviewer #3 (Remarks to the Author):

This manuscript reports very ambitious and important results on a Class C GPCR. Two aspects are important; relatively little is known about Class C GPCR mechanisms in comparison to Class A, or even Class B, and the importance of dimerization and its meaning for the mechanism are still relatively little studied. Thus, the work is certainly worthy of Nature Commun.

The simulations have, as far as I can judge from the manuscript, been conducted carefully and correctly. However, I do have some doubts about whether everything really is converged.

Although the simulations are quite long, even by today's standards, our experience suggests that GPCR simulations in membranes often display an induction period of approximately 500 ns before sometimes quite significant conformational changes occur. We attribute this phenomenon to a bias of the force fields towards X-ray structures, making them local minima from which the simulation must escape. Our experience is based on the AMBER force field but this is generally "faster" than the CHARMM force field use here. I would therefore like to see evidence that the equilibration simulations really are equilibrated.

The same is true of the metadynamics. One of the advantages of this technique is that convergence can be checked (and demonstrated) by plotting the free-energy profile as a function of simulation time. Convergence should certainly be demonstrated.

Reviewer #1 Remarks to the Author, reproduced below with the Author's responses in bold face

The GABA_BR is a class C GPCR whose active conformation has recently been solved, demonstrating a significant conformational shift from the inactive or partially activated structures. The authors used MD and metaD simulations to predict the mechanism of activation for this heterodimeric receptor. They discuss a novel mechanism of activation and opening of the G protein to exchange GDP for GTP. This may be common mechanism amongst class C GPCRs.

The authors used MD and metaD simulations to predict the conformational changes that the GABA_BR undergoes during activation and G protein coupling. This is the first proposed mechanism for this in Class C GPCRs and represents a significantly different mechanism from that of Class A GPCRs. The predicted mechanism explains the differences seen in the various cryo-EM structures of the GABA_BR in different activation states.

Thank you for the valuable comments. Please find below our responses.

Major Comments:

1. The methods described in this manuscript can be used to predict the structural rearrangements of different transmembrane proteins and their signaling partners. Specifically, now that the interactions that govern G protein coupling and activation for this heterodimeric receptor have been mapped, it is possible to design novel agonists/antagonists/allosteric modifiers to trap the receptor in different activation states. It also paves the way for similar studies in other Class C GPCRs. A discussion explicitly suggesting how one could trap the different conformations experimentally based on their simulation results will be critical for impact in the field.

Based on our predictions, the G protein (the terminus of the G α subunit; F354G α 5) interacts first with R577 in TM3 of GB2 as it approaches to the receptor, and then further moves toward ICL1 for the fully activated state. Thus, one may be able to capture the intermediate state predicted in this study by mutating the positive residues in ICL1, preventing formation of the final activated structure. In fact, mutation of either K510A or K513A showed that abolishing either of these residues does not significantly change the receptor activity (ref. 17 in the text). These experimental results are consistent with our structure since there are close enough that either could make a salt bridge. However we predict that abolishing both K510 and K513 residues would prevent proceeding to the final state, leading to trapping of the intermediate state. We added a sentence to discuss this point.

In the revised manuscript (page 9, lines 7-12)

“This also suggests that this pre-activated state may be able to be experimentally captured by mutating the positive residues in ICL1, although it is not clear how stable such intermediates would be. Experimental results showed that mutation of either K510A or K513A did not significantly change the receptor activity.¹⁷ Even so, we expect that abolishing

both lysine residues could trap the system in the intermediate state, preventing the complex from proceeding to the final state.”

2. The significance of the elucidation of the meta-stable state of mGABA_B is striking; however, there is no experimental evidence that validates the in silico findings. The authors could reference Liu et al 2019 (insert reference to Kobilka’s b-AR-Gs fusion protein) and perhaps replicate the results using a Gi fragment-mGABA_B fusion protein. Alternatively, they could consider using hydrogen-deuterium exchange to see if there is exchange between the residues implicated in these interactions, or perhaps label specific residues and use NMR to study the meta-stable state. These maybe outside the scope of the manuscript but an explicit set of hypotheses they can deduce and discussion of experimental design that will allow others to prove or disprove the proposed mechanism will be critical for consideration.

The paper (Liu et al 2019) reported an intermediate state focused on the interaction between the β 2AR receptor and the G α 5 helix of the Gs protein, where the G α 5 helix was considered to interact in stages from its first contact to the receptor to the final fully activated structure. Moreover, this paper reported that the charged or aromatic residues of the G α 5 helix, such as R389 and E392, play an important role during this process, which agrees with our proposed mechanism (corresponding to D350 and F354COO⁻ (terminal carboxylate) of the G α 5 helix of the Gi protein). As we mentioned above, mutation of the positive residues in ICL1 could lead to identification of such intermediate states, providing useful information for understanding the entire activation process. We now cite this paper and added discussion about this point.

In the revised manuscript (page 9, lines 13-16)

“For β 2-adrenergic receptor with the carboxyl terminal 14 amino acids from G α s protein, an intermediate state was reported,⁴¹ where the charged residues of the G α 5 helix (R389 and E392) are considered to play an important role in complex formation and they change the interactions with the receptor from its first contact to the receptor to the final fully activated structure.”

3. As presented, there is little novelty in the MD simulations by themselves, leaving only the uniqueness of the activation mechanism. Some experimental data validating the meta-stable state will strengthen the results. Otherwise, the study represents an important prediction and could be published as such upon appropriate revision. Irrespective of this, it will be important to discuss the findings by comparing and contrastic what has been discussed in these recent papers: <https://pubmed.ncbi.nlm.nih.gov/35296853/> <https://pubmed.ncbi.nlm.nih.gov/34759375/>

All GPCRs show significant conformational change upon activation, and the movement and rotation of TM6 is a universal feature of activation across the GPCR superfamily. However, the activation mechanism for class C GPCRs differs significantly from other class GPCRs, where the lack of TM6 movement (and also lack of conserved prolines in TM5, 6, and 7, unlike class A) is one of its unique features. We added discussions about the comparison of

conformational changes upon activation for other class GPCRs, particularly focused on TM6.

In the revised manuscript (page 7, lines 1-12)

“GPCR shows significant conformational change upon activation, with the movement and rotation of TM6 that is a universal feature of activation across the GPCR superfamily, leading to accommodation of the G α subunit. However, this TM6 movement displays large mechanistic differences across classes.³⁴ In addition to class A, it has been reported that a class B GPCR shows an significant outward movement, however in a different way; the TM6 helix is disrupted and a sharp kink is formed.³⁵ Also, a class D GPCR showed a unique feature, the outward movement of the extracellular side and the inward movement of the intracellular side of TM6.³⁶ For class F, it has been reported an outward movement of TM6 similar to class A.³⁷ These indicate that TM6 is a universal switch for activation in each class, however it undergoes different rearrangements in each class. Because of such distinct mechanisms in each class, to understand the activation mechanism at the level of helices/residue is important, which provides insights into rational design of experiments and eventually drug discovery.”

Other Comments

1. The text is difficult to read due to the excessive use of acronyms. If a word or phrase is used fewer than 3 times, it does not need to be abbreviated (ie, CRD, Cysteine Rich Domain is only used twice in the entire document).

We removed unnecessary or used less than 3 times acronyms in the text; CaS, TAS1, and CRD.

2. The methods section does not state the length of time for the individual MD production runs. This information is necessary to assess the confidence in sampling sufficient conformational space within the MD simulation.

We added the simulation times for MD and metaD simulations we performed in Methods section.

In the revised manuscript (page 12, lines 3-5)

“The total simulation times for GABA_BR heterodimer with and without GiP were 200 ns and 500 ns, respectively.

The MD simulation for the GABA_BR heterodimer without PAM was performed for 800 ns on Anton2 in the NPT ensemble with 1 bar and 310 K.”

3. There are several uses of italics that seem unnecessary, pg 2 line 3, 6, 14, 36, 38; pg 3 line 16

We corrected the unnecessary italics the reviewer pointed out.

4. The text needs careful reading, as there are places where articles (page 2 line 2: ...is one of <the> principal inhibitory...) or other words have been left out

We corrected the sentence.

5. Page 7 line 26 “It is well known...” the reviewer feels this statement requires a citation

We added a citation about the statement (Zhou, Q. et al. Common activation mechanism of class A GPCRs. eLife 8, e50279 (2019)).

6. Page 8 line 25 the sentence that begins with “Hence,...” Hence is not the appropriate word, as the statement does not flow logically from the previous sentence.

We corrected the word from “Hence” to “Moreover”.

7. Page 9 line 23 “...helix rotates and translates into the cytoplasmic side...” translates does not seem like the best verb here

We corrected the word from “translates into” to “moves toward”.

Reviewer #2 Remarks to the Author, reproduced below with the Author's responses in bold face

In the present study molecular dynamics and molecular metadynamics are used to study the molecular mechanisms of Gi-protein interactions and activation of the GABA-B receptor. The entire activation process following agonist binding is not studied, but focus is on interactions between the receptor and the Gi-protein. The study is well designed and performed and necessary information is given in the methods part for reproducing the results. To my opinion, the manuscript can be accepted for publication after some changes.

Thank you for the favorable comments. Please find below our responses.

Comments/suggestions to the authors:

1. A main observation from these studies was that the Gi-protein plays an active role in the conformational changes of the receptor during activation leading to a rearrangement of TM3-4-5 of GB2 to form an active TMD conformation. However, this seems to be contradictory to the cryo-EM study by Chen and co-workers from 2021 (ref. 17 in the manuscript). They write that the cryo-EM structure of the agonist-PAM-Gi-bound GABA_B structure was almost identical to that observed with the agonist and PAM without Gi, indicating that Gi had no effect on the conformation of the receptor. Baclofen and BHFF were also used in the cryo-EM study. The authors should comment on that in the results and discussion section.

As the reviewer pointed out and the authors of ref. 17 mentioned in their paper, the overall structures of GABA_BR TMD are quite similar for the inactive and active conformations. However, there are subtle but clear conformational changes in the TMD, particularly, TM3-4-5 upon activation as we described in the text. Figure 3f in fact shows the direct structural comparison of TM3-4-5 in between inactive and active states. To make this point clearer, we added a sentence.

In the revised manuscript (page 8, lines 6-8)

“Although each monomer of GABA_BR shares a quite similar conformation for the inactive and active states, TM3-4-5 shows subtle but clear conformational changes upon activation.”

2. Not all readers are familiar with metadynamics, and the author should therefore include a short description of metadynamics in the introduction, and why it is convenient to use that for studying the coupling mechanism.

We added the brief description about metadynamics into Introduction section.

In the revised manuscript (page 3, lines 4-8)

“MetaD is used to follow the free energy profile between states when the barrier is too high to be overcome using regular MD over a reasonable time scale. As the metaD proceeds to convergence, this process fills the potential energy for a suitably chosen set of collective variables, where the free energy surface is obtained by backing out the added terms.”

3. In the first paragraph of the introduction, they mention that GABA-B as a drug target in neurological and psychiatric disorders and give three references (4-6). Reference 6 is connected to cancer?

In addition to its neurological functions in the central nervous system, it has been reported that GABA_BR is also related to cancer. We added a sentence in the first paragraph.

In the revised manuscript (page 2, lines 10)

“Moreover, it has been also reported that GABA_BR is involved in tumour development.”⁶”

4. In my version, the methods part is inserted into the middle of the reference list before all references to the methods part. Something wrong with the formatting?

We corrected the manuscript format.

5. The cryo-EM structures show a PAM binding site in between the TMD interfaces, and that the PAM stabilizes an active TMD conformation. MD simulations showed that the active TMD conformation with the TM6/TM6 interface is stable without BHFF present. AgoPAMs can activate the receptor without agonist present, indicating that agoPAMs can induce active receptor conformations. Based on your simulations and available cryo-EM structures can you comment/speculate on the mechanisms of GABA-B agoPAMs.

Ago-PAM leads to intrinsic agonist activity just as does an agonist. The BHFF used in this study is one of ago-PAMs and thus it is capable of causing the receptor activity by itself. Based on this binding site at the dimeric interface, we speculate that BHFF increases the rate of the GP engagement by stabilizing the active TM6-TM6 interface even when agonist is not present. We added discussion about ago-PAM into the manuscript.

In the revised manuscript (page 4, lines 3-10)

“PAMs typically potentiate the receptor activation induced by an agonist, however some PAMs also exhibit intrinsic agonist activity even when the agonist is not present. These are called ago-PAMs.²¹ It has been reported that ago-PAMs can cause the conformational changes in GB1-VFT and TMD observed upon agonist activation.^{22,23} BHFF is a known ago-PAM, which is capable of causing the receptor activity alone.²³ Since BHFF binds to the dimeric interface, it is speculated that it increases the chance for the GP engagement by stabilizing the active TM6-TM6 interface even without the agonist. In fact, it is important to develop such allosteric compounds because they can modulate the effect of orthosteric ligands to increase desirable therapeutic effects.²¹”

Reviewer #3 Remarks to the Author, reproduced below with the Author's responses in bold face

This manuscript reports very ambitious and important results on a Class C GPCR. Two aspects are important; relatively little is known about Class C GPCR mechanisms in comparison to Class A, or even Class B, and the importance of dimerization and its meaning for the mechanism are still relatively little studied. Thus, the work is certainly worthy of Nature Commun.

Thank you for the positive comments. Please find below our responses.

The simulations have, as far as I can judge from the manuscript, been conducted carefully and correctly. However, I do have some doubts about whether everything really is converged.

Although the simulations are quite long, even by today's standards, our experience suggests that GPCR simulations in membranes often display an induction period of approximately 500 ns before sometimes quite significant conformational changes occur. We attribute this phenomenon to a bias of the force fields towards X-ray structures, making them local minima from which the simulation must escape. Our experience is based on the AMBER force field but this is generally "faster" than the CHARMM force field use here. I would therefore like to see evidence that the equilibration simulations really are equilibrated.

As the reviewer pointed out, to check the convergence or equilibration of the simulations is very important. Also, some papers have reported that significant and interesting conformational changes from μ s order MD simulations (e.g. Dror et al. Science 2015; Latorraca et al. Nature 2018; Suomivuori et al. Science 2020). In our study, although the simulation time for a single MD run was less than μ s, we believe that our simulations reached in equilibration particularly for the TMD region focused on in our study.

For the fully activated state, we started with the well-defined cryo-EM structures (several cryo-EM studies showed almost identical conformations; refs. 9, 13, 15-17 in the text). We found that this conformation was stable and maintained during the simulations particularly for the TMD region. To make this point clearer, we included additional RMSD plots calculated with respect to the initial structure of the MD simulations (Fig. S1 and S4e).

For the intermediate state, a longer MD simulation ($\sim\mu$ s or even longer) could possibly lead to a conformational change toward an more activated structure that might include the effect of long-range allosteric interactions. This would be an interesting point to explore and a worthy study to complete the entire activation process. However, we consider this to be beyond the scope of this study focused on the initial interactions between the receptor and the G protein as an intermediate state. To make this point clear, we added sentences.

In the revised manuscript (page 10, lines 11-15)

“Since the pre-activated structure is an intermediate state, a longer MD simulation (~ μ s or even longer) may lead to additional conformational changes toward the activated structure, such as the G α opening and GDP exposure. Instead, metaD simulation can reduce the computational cost and estimate the energy along the reaction path.”

The same is true of the metadynamics. One of the advantages of this technique is that convergence can be checked (and demonstrated) by plotting the free-energy profile as a function of simulation time. Convergence should certainly be demonstrated.

As the reviewer suggested, we provide plots of the free energy profile as a function of simulation time for each metaD simulation in the Supplementary Information (Fig. S7 and S8), and added a sentence to make it clear.

In the revised manuscript (page 13, lines 15-17)

“Free energy profiles as a function of simulation time were plotted to check the convergence of metaD simulations (Supplementary Fig. 7 and Fig. 8).”

REVIEWERS' COMMENTS

Reviewer #1 (Remarks to the Author):

The authors have adequately addressed the issues raised by this referee. I am happy with this version of the manuscript.

Reviewer #2 (Remarks to the Author):

The authors have addressed all my concerns regarding the manuscript and revised the manuscript accordingly. To my opinion, the manuscript is ready for publication.

Reviewer #3 (Remarks to the Author):

The authors have taken my comments (and those of the other reviewers) into account, so that the manuscript can now be published.